# Agar-Dilution Is Comparable to Broth Dilution for MIC Determination in *Streptococcus agalactiae*

**DOI:** 10.3390/antibiotics14020156

**Published:** 2025-02-05

**Authors:** Edward A. R. Portal, Caitlin Farley, Teresa Iannetelli, Juliana Coelho, Androulla Efstratiou, Stephen D. Bentley, Victoria J. Chalker, Owen B. Spiller

**Affiliations:** 1Division of Infection and Immunity, Department of Medical Microbiology, University Hospital of Wales, Cardiff University, Cardiff CF14 4XN, UK; edward.portal@biology.ox.ac.uk (E.A.R.P.); teresa.iannetelli@biology.ox.ac.uk (T.I.); 2Department of Biology, Ineos Oxford Institute of Antimicrobial Research, University of Oxford, Oxford OX1 3RE, UK; 3Reference Microbiology Division, United Kingdom Health Security Agency, London NW9 5EQ, UK; juliana.coelho@ukhsa.gov.uk (J.C.); androulla.efstratiou@ukhsa.gov.uk (A.E.);; 4Parasites and Microbes, Wellcome Sanger Institute, Cambridge CB10 1SA, UK; sdb@sanger.ac.uk; 5Office of the Chief Scientific Officer for the UK, London SE1 8UG, UK

**Keywords:** *Streptococcus agalactiae*, GBS, antimicrobial resistance, antimicrobial susceptibility testing, method validation

## Abstract

Background: *Streptococcus agalactiae* (Group B Streptococcus, GBS) is a leading cause of neonatal sepsis in high-income countries. While intrapartum antibiotic screening reduces this risk, increasing resistance to macrolides and lincosamides in Europe since the 1990s has limited therapeutic options for penicillin-allergic patients. Reports of reduced beta-lactam susceptibility in GBS further emphasise the need for robust antimicrobial resistance (AMR) surveillance. However, broth microdilution (BMD) methods are unsuitable for large-scale antimicrobial susceptibility testing (AST). Objective: To demonstrate that agar-dilution AST provides equivalent results to broth dilution methods, with superior capacity for high-throughput screening. Methods: Agar-dilution and microdilution AST methods were compared using a panel of 24 characterised susceptible and resistant GBS strains for benzylpenicillin, chloramphenicol, clindamycin, erythromycin, gentamicin, levofloxacin, tetracycline, and vancomycin. Minimum inhibitory concentration (MIC) agreements were evaluated, and resistance profile correlations were assessed using Cohen’s kappa values. Results: Agar-dilution demonstrated >90% agreement with BMD MIC for most antimicrobials, except vancomycin (87.5%), erythromycin (83.33%), and tetracycline (52.78%). Cohen’s kappa values indicated strong agreement (0.88–1.00) for resistance determination. Agar-dilution avoided “trailing growth” issues associated with BMD and facilitated easier detection of non-GBS contaminants. Conclusions: Agar-dilution is a valid method for high-throughput AMR surveillance of retrospective cohorts (96 isolates per plate) and is critical for identifying emerging GBS resistance trends and informing therapeutic guidelines. However, due to the large number of plates required per antimicrobial, it is impractical for routine clinical diagnostics.

## 1. Introduction

*Streptococcus agalactiae*, also known as Group B *Streptococcus* (GBS), is a leading cause of neonatal and infant infections [1]. In neonates, GBS, alongside *Escherichia coli*, is the most common cause of culture-confirmed sepsis in high-income countries (HICs) [2], with a case fatality rate of 3.4% in term births and 3.7% in cases of extreme prematurity [3]. GBS meningitis is associated with high mortality, and neurological impairments have been documented in 32–44% of surviving neonates [4]. Treating GBS infections promptly and effectively is crucial to reducing mortality and long-term complications [5,6]. The National Institute for Health and Care Excellence (NICE) recommends initiating prophylactic antibiotic treatment within one hour of suspected sepsis diagnosis.

In the UK, guidance from the Royal College of Obstetricians & Gynaecologists (Green-top Guidelines [7]) advises against universal GBS screening but recommends providing pregnant women with informational leaflets. Antibiotic therapy should be initiated in cases where GBS was detected in a previous pregnancy, during the current pregnancy, or in the presence of pyrexia during labour. Benzylpenicillin is the first-line treatment, with cephalosporins or vancomycin as alternatives for penicillin-allergic patients [8]. Neonatal treatment typically includes gentamicin combined with either benzylpenicillin or ampicillin. Erythromycin and clindamycin are not recommended [9,10].

Antibiotic resistance in GBS is a growing concern globally. The gold standard susceptibility assay is broth microdilution (BMD), as per EUCAST and ISO 20776-1 standards [11,12,13]. However, BMD can be problematic for GBS due to the need for lysed horse blood supplements, which complicate turbidity readings [12]. The use of automated systems (e.g., VITEK2, Phoenix, Microscan WalkAway, etc.) is more common in large clinical microbiology laboratories, and they perform well on GBS. The use of Kirby–Bauer disc diffusion on solid agar is easier to perform in small-scale settings and non-GBS bacterial contaminants are easier to identify on agar. MIC determination using serial antibiotic agar-dilution is not a listed recommended method in EUCAST or CLSI (Clinical and Laboratory Standards Institute) GBS susceptibility testing guidelines. However, it has been used in Belgium and Japan with more than 200 GBS isolates [2,14].

This study compares BMD and agar-dilution methodologies using twenty-four GBS strains from a previously published cohort of UK invasive isolates of known antibiotic susceptibility profiles, correlated to whole genome sequence analysis that established the underlying resistance mediating antimicrobial resistance genes (ARGs) or somatic mutations (except for penicillin and vancomycin) [15]. This panel also included a subset of characterised wild type strains that were fully susceptible to the panel of antimicrobials investigated in this study, including tetracycline. The purpose of this study was to demonstrate that determining MICs by agar-dilution gives equivalent results to broth microdilution (BMD), readily being capable of separating isolates carrying predetermined resistance determinants from susceptible controls.

## 2. Results

Details of the twenty-four GBS strains utilised including NCTC (National Collection of Type Culture) numbers, ARGs, serotype, and sequence type are listed in Table 1. Quality control strains *Streptococcus pneumoniae* ATCC (American Typed Culture Collection) 700677 (resistant to erythromycin, penicillin, and tetracycline) and *Streptococcus pneumoniae* NCTC12977 (susceptible to all) were also run in parallel, as per EUCAST guidelines [12]. All isolates were initially cultured on Colombia Horse Blood Agar (Oxoid, Basingstoke, UK) at 37 °C under ambient O_2_ concentrations and single colonies were re-suspended in 3 mL sterile 0.85% saline (ThermoFisher Scientific, Abingdon, UK) at a concentration of 0.5 McFarland (1.5 × 10^8^ CFU/mL).

Determining minimum inhibitory concentrations (MICs) by assessing bacterial growth turbidity in Mueller–Hinton Fastidious (MH-F) broth was more challenging compared to using standard Mueller–Hinton (MH) broth (e.g., for *E. coli*) due to “trailing growth” [12], a phenomenon in which the turbidity cut-off for bacterial growth inhibition is unclear, as we consistently observed with MIC determinations for erythromycin. Despite this, good MIC concordance was observed between the agar-dilution and BMD methods (Figure 1).

Using EUCAST established resistance thresholds (Table 1), all strains positive for *tet*(L), *tet*(M), and/or *tet*(O) were resistant to tetracycline and isolates carrying mutations *gyrA* S81L and/or *parC* S79F were resistant to levofloxacin (Figure 1). Isolates carrying *mef*(A) and *msr*(D) as the sole macrolide resistance genes were consistently identified as resistant to erythromycin (MICs 2–4 mg/L) by both methods. Recently, EUCAST guidelines have changed for chloramphenicol, where 8 mg/L, which was previously set as the threshold for “resistant”, has been altered to “IE” (Insufficient Evidence) in the EUCAST Clinical Breakpoint Tables v. 14.0; however, all strains positive for *cat(Q)* or *cat(C194)* were above the previous threshold (Figure 1). While the lowest gentamicin MIC for any GBS isolate is well above a reasonable pharmacokinetic/pharmacodynamic (PK/PD) threshold for clinical intervention (0.5 mg/L), the single isolate carrying *aac(6′)-aph(2″)* exhibited a gentamicin MIC > 128 mg/L, relative to 16–32 mg/L observed for all other isolates by both methods.

Data for GBS isolates carrying *erm*(A) and *erm*(B) methylases showed less consistency: three *erm*(A)-carrying isolates gave an MIC of 0.25 mg/L (just below the resistance threshold) in one of three BMD replicates, while a single *erm*(B)-carrying isolate (PHEGBS0738) gave an MIC of 0.25 mg/L for two of three BMD replicates and one of three agar-dilution replicates. Although methylases often require macrolide induction for clindamycin resistance, one of seven *erm*(A)-carrying isolates and six of seven *erm*(B)-carrying isolates displayed clindamycin resistance in both methods. An isolate carrying *lsa*(E) (NCTC14907) was consistently susceptible to clindamycin, with an MIC of 0.25 mg/L for two of three BMD replicates. Additionally, an isolate carrying both *lnu*(C) and *erm*(A) (NCTC14903) was consistently clindamycin-susceptible by both methods.

Strain-matched MIC concordance analysis between BMD and agar-dilution (Table 2) demonstrated that most results were within one dilution, regardless of the method. This establishes that agar-dilution is comparable to BMD for MIC determination, with concordance rates mostly ranging from 83–100% across antibiotics and strains tested. Tetracycline was an exception, with only 52% concordance and greater variability between replicates. Notably, tetracycline MICs consistently increased with extended incubation, a phenomenon not observed for other antibiotics.

Evaluation of the degree of agreement across agar-dilution and BMD methods for defining isolate susceptibility profiles (Table 3) revealed near-perfect agreement (kappa value > 0.9) for chloramphenicol, clindamycin, erythromycin, levofloxacin, and tetracycline. Despite the poor concordance of individual tetracycline (52.78%) MICs, a high agreement and susceptibility concordance (100%) was obtained due MIC variability not crossing the susceptibility threshold. Erythromycin, despite having a greater MIC concordance (83.33%), only obtained a moderate agreement due to variability across the MIC threshold. No kappa values could be calculated for benzylpenicillin, gentamicin, and vancomycin due to a complete agreement between methods categorising all isolates into one susceptibility category.

## 3. Discussion

With a few exceptions, EUCAST recommends the use of the broth microdilution reference method as the AST gold standard, including fastidious organisms, using MH-F [12]. Similarly, CLSI also indicate that agar-dilution has not been internally performed or reviewed [11]. However, interrogation of large cohorts against multiple antibiotics by BMD is not feasible, contamination is easier to identify on agar, and avoids “trailing growth” [11] (a well-known phenomenon that complicates BMD determination MICs). There are many reports in the literature determining MICs using agar-dilution [2,14,16,17], but to date a systematic comparison of BMD and agar-dilution concordance for GBS has not been performed, as Reynolds et al., have performed for *S. pneumoniae* [18].

Amsler et al., [17] reported a consistent 2-fold (on average) lower MIC for agar-dilution relative to BMD for a combined subgroup of 21 GBS and 18 *Streptococcus pyogenes*, but with a 98.2% agreement within ±2 log2 dilutions. We did not observe this skewing in our study, except for tetracycline which was also particularly variable, with only 52.78% agreement within ±1 log2 dilution. Greater than 90% agreement, within ±1 log2 dilution, was observed for all other antimicrobials except erythromycin (83.33%) and vancomycin (87.5%).

In this study, the variability in tetracycline MICs remains unexplained, although others have reported that doxycycline degrades rapidly in solution over time [19]. The pronounced variability we found for tetracycline MICs remains unexplained, especially as it was not subject to the “trailing growth” we observed for erythromycin in BMD. While Reynolds et al., [18] indicated more variance between BMD and agar-dilution for tetracycline than erythromycin, clindamycin, and levofloxacin, they still found 98.9% agreement when the comparison was extended to ±2 log2 dilutions compared to our 80.5% agreement for the same range.

Tetracycline is well known to be more labile, and we have previously reported a slow but significant increase in doxycycline or tetracycline MICs with incubation time using both agar-dilution [20] and broth dilution [21] methods for determining AST for *Legionella pneumophila*. This increase was not observed for other antimicrobials tested. It has been reported that tetracycline family binding to the ribosome (target inhibition site) requires magnesium ions and that Fe^3+^ ions oxidize and accelerate the degradation of tetracycline [22,23]. We have found no alteration to tetracycline MICs when altering several variables (e.g., sequestering iron ions, adding/removing magnesium, and adding/removing serum proteins) compared to BMD tetracycline MICs run in parallel with our standardized broth for *Legionella pneumophila*. Therefore, there does not appear to be any recommendation that would reduce the inherent variability in tetracycline AST, which is the primary factor in discordant comparisons for this method. It is possible that the transient elevation of temperature to 50 °C during agar-dilution or interactions with the agar itself is responsible for the discordance.

Tetracycline overuse of the 1960s left a legacy of most (>90%) GBS carrying tetracycline resistance genes [24]; therefore, screening for tetracycline resistance is unlikely to yield clinically meaningful results. Agar-dilution remains suitable for MIC determination for other therapeutically relevant antimicrobials. This is particularly important when screening large cohorts of GBS isolates in antimicrobial resistance surveillance studies. The method outlined here, using a multipin inoculator, is ideally suited for the high-throughput retrospective analysis of large-archived cohorts to evaluate resistance trends. However, as ECOFF thresholds are established for most of these antimicrobials, a modified version for routine screening could be utilised: a single antimicrobial-free plate used as a growth control combined with a single plate containing the threshold concentration for each antimicrobial to be tested (e.g., 0.5 mg/L benzylpenicillin, 2 mg/L levofloxacin, etc.). Using smaller plates and inoculating one microlitre of the McFarland suspension with a pipette tip for routine screening would also eliminate the need for a multipin inoculator. We recommended using threshold concentrations for resistance screening in *Ureaplasma* spp. and *Mycoplasma hominis* in 2009 [25], which has since been adopted (using the threshold concentration and one concentration below for three antimicrobials) and is now commonplace in commercial clinical screening assays for these bacterial species [26,27]. Routine screening in this fashion would also have the advantage that stocks of threshold screening plates can be prepared monthly and stored more easily than 96-well plates containing antimicrobial dilutions in broth.

Gentamicin demonstrated perfect concordance across methods, largely because the presence of the gene *aac(6′)-aph(2″)* elevates the MIC to the top of the measured range. Bactericidal antimicrobials did not exhibit higher concordance than bacteriostatic agents (Table 2); however, *aac(6′)-aph(2″)* was the only resistance gene tested that physically alters the antimicrobial (by adding phosphate and acetyl groups) as its mechanism of action, rather than modifying the host antimicrobial binding target.

Gentamicin is bactericidal and therefore does not require consistently high systemic levels to be effective. However, the average gentamicin MIC for GBS is higher than the anticipated peak serum levels (Cmax) of 5–10 mg/L achieved with multiple daily dosing regimens [28]. There is, however, in vitro evidence that gentamicin exhibits synergistic effects in killing GBS when combined with penicillin, which is commonly co-administered with gentamicin [29].

In conclusion, a direct comparison of MIC determination between agar and broth dilution methods is presented, using a defined cohort of GBS isolates with characterised resistance genes or mutations. This study found excellent concordance for seven out of the eight antibiotics tested. Agar-dilution may be utilised for routine clinical laboratory use, particularly in settings with high sample loads, or when modified to screen growth on plates containing resistance threshold concentrations of antimicrobials. Further studies to assess its clinical value could be beneficial. Additionally, it would be prudent to use agar-dilution for retrospective batch analysis in AMR surveillance, as it allows large-scale screening for multiple antibiotics, contributing to the evidence base for resistance trends and informing policy on first- and second-line therapeutics.

## 4. Materials and Methods

Antimicrobial sensitivity testing was compared using Muller Hinton Fastidious (MH-F) broth (BD, Franklin Lakes, NJ, USA) in 96-well plates and MH-F agar (Neogen, Lansing, MI, USA) in 90 mm petri dishes, both supplemented with 5% lysed horse blood (TCS Biosciences, Buckingham, UK) and β-nicotinamide adenine dinucleotide as per EUCAST guidelines. [10]. MICs were determined for benzylpenicillin, chloramphenicol, clindamycin, erythromycin, gentamicin, levofloxacin, tetracycline, and vancomycin (Sigma-Aldrich, Dorset, UK) at concentration ranges between 0.008–128 mg/L. All data points were obtained in triplicate.

### 4.1. Agar Dilution Method

For solid agar-dilution, antibiotic stocks were prepared at 100 mg/mL in either dimethyl sulfoxide (DMSO) or water and diluted into three starting stocks of 2560, 80, and 2.5 mg/L. These were then aliquoted into individually labelled 30 mL universal containers (UCs) to achieve the full serial dilution as shown in Table 4.

Four-hundred and seventy-five mL of Muller–Hinton (MH) agar (Sigma-Aldrich, UK) was autoclaved and equilibrated to 50 °C. Twenty-five mL (final concentration 5%) of lysed horse blood was aliquoted from a 500 mL bottle stored at 4 °C (TCS bioscience, UK) and brought to room temperature. β-NAD was supplied as 20 mg vials (Sigma-Aldrich, UK), dissolved in 1000 µL of ddH_2_O, and sterilised through a 0.22 µM filter. Per 500 mL of MH-F, 500 µL of dissolved β-NAD (20 mg/mL) was then mixed with lysed horse blood into the 475 mL of 50 °C MH agar. Then, 20 mL of the above mix was poured into individually labelled 30 mL UCs containing antimicrobials in quantities described in Table 4 to achieve the correct dilution series. Then, it was gently mixed and poured into round 90 mm agar plates.

Once plates were solid and allowed to dry thoroughly (15 min), a 0.5 McFarland suspension was made for each bacterial isolate to be tested and a 1:10 dilution aliquoted into a sterile 96-well plate. Once all isolates were prepared in the inoculation plate, it was placed in a Mast Uri^®^ Dot multipin inoculator (1 μL pin volume) (Mast Group Ltd., Liverpool, UK) and the prepared agar-dilution plates were “stamped” in ascending order (starting with an antimicrobial-free growth control plate). Inoculated plates were allowed to dry at room temperature in the biological safety cabinet for 15–30 min (to avoid streaking when turned), inverted, and incubated for 18 h at 37 °C, after which results were read.

### 4.2. Broth Microdilution Method

For broth microdilution (BMD), 0.1 mL of MH-F broth was added to each well and 0.1 mL containing 256 mg/L antibiotic was added to the first row, mixed, and then serially transferred across the plate to give a gradient of 128–0.008 mg/L, with the exception of the last two rows (penultimate row represents antibiotic free growth control and last row left for bacteria-free MH-F broth sterility control). Plates were incubated at 37 °C overnight without CO_2_ and BMD results were read using a light box as per EUCAST guidelines. All experiments were conducted in triplicate. MIC thresholds for resistance are listed in Table 5.

### 4.3. Statistical Analysis

Data were analysed by overall percentage concordance, fold-change in MICs between repeats, and by Cohen’s kappa to measure agreement of resistance profiles. The weighted kappa statistic was calculated using the Quantify agreement with the kappa online calculator (www.graphpad.com/quickcalcs/kappa1/?K=3; accessed on 10 August 2022). Interpretation of the kappa statistic followed the categories defined by Landis and Koch [30].

### 4.4. Ethical Approval

Ethical approval was not required as only NCTC-or ATCC-deposited type strains were used.

## 5. Conclusions

We showed an acceptable level of concordance between the agar-dilution and broth microdilution methods to establish the former as an acceptable and valid alternative to the latter method for AST. Agar-dilution is a superior method for high-throughput AMR surveillance of retrospective cohorts as up to 12 different antimicrobials can easily be performed on 96 isolates per inoculation plate. We recognise that agar-dilution would be impractical for routine clinical diagnostic use, due to the requirement of individual plates to establish a range of antimicrobials; however, agar-dilution would be the method of choice where evaluation of retrospective cohorts would lend itself to identifying emerging GBS resistance trends and informing therapeutic guidelines.

## Figures and Tables

**Figure 1 antibiotics-14-00156-f001:**
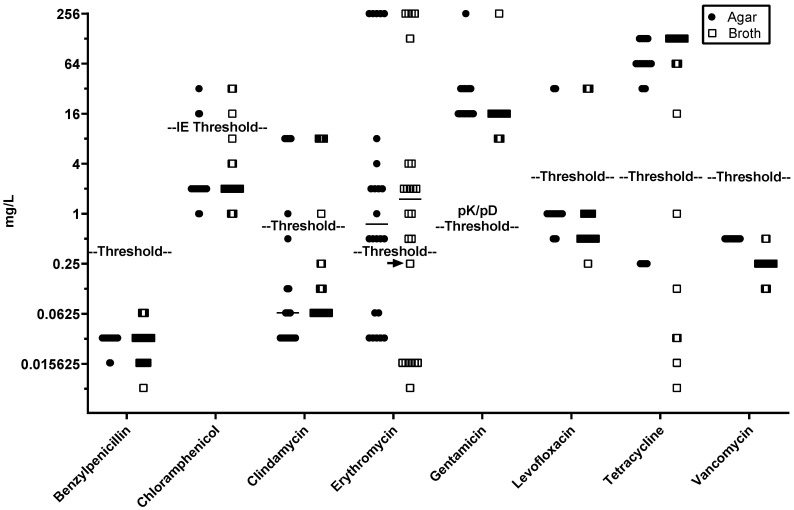
MIC determination for 24 GBS isolates using agar-dilution (closed circle) or BMD (open square) methods. Thresholds for resistance cut-off are shown (gentamicin details pK/pD values as EUCAST breakpoints are not available, and V14.0 EUCAST guidelines has downgraded the chloramphenicol threshold to IE for insufficient evidence). Antibiotic test range was 0.008–128 mg/L except for clindamycin (0.008–> 4 mg/L as indicated on graph). The arrows indicate *erm*A isolates that fell below the accepted resistance threshold concentration.

**Table 1 antibiotics-14-00156-t001:** Isolates used for agar and broth comparison.

NCTC Number	Resistance Genes	Original Reference	Serotype	Sequence Type
14894	*erm*(A) *tet*(M)	PHEGBS0044	Ia	ST23
14895	*erm*(A) *tet*(M)	PHEGBS0066	III	ST17
14896	*mef*(A) *msr*(D) *tet*(M)	PHEGBS0067	Ia	ST23
14897	*mef*(A) *msr*(D) *tet*(M)	PHEGBS0070	Ia	ST23
14898	*lsa*(C) *erm*(B) *tet*(M) *tet*(O)	PHEGBS0071	III	ST19
14899	*erm*(A) *tet*(M)	PHEGBS0082	V	ST1
14900	*erm*(A) *tet*(O)	PHEGBS0091	II	ST12
14901	*erm*(A) *tet*(M)	PHEGBS0098	V	ST1
14902	*erm*(B) *tet*(M)	PHEGBS0128	V	ST1
14903	*aac*(6′) *aph*(2”), *erm*(A) *lnu*(C) *tet*(M)	PHEGBS0139	V	ST19
14904	None	PHEGBS0408	II	ST28
14905	None	PHEGBS0446	VI	ST1
14906	None	PHEGBS0491	II	ST12
14907	*lsa*(C) *tet*(M)	PHEGBS0511	IV	ST297
14908	*ant*(6-Ia) *aph*(3′-III), *aadE*, *erm*(B) *msr*(D) *mef*(A) *tet*(O)	PHEGBS0577	III	ST17
14909	*tet*(L) *tet*(M)	PHEGBS0586	II	ST652
14910	*None*	PHEGBS0592	IX	ST130
14911	*erm*(A) *msr*(D) *mef*(A) *tet*(M) *catQ*	PHEGBS0608	V	ST19
14912	*erm*(B) *tet*(O) *tet*(M)	PHEGBS0624	II	ST28
14913	*erm*(B) *ant*(6-Ia) *aph*(3′-III) *tet*(S)	PHEGBS0662	VI	ST1
14914	*erm*(B) *tet*(M) *ant*(6-Ia) *aph*(3′-III) *cat*(C194)	PHEGBS0738	V	ST19
14915	*erm*(B) *msr*(D) *mef*(A) *tet*(O) *ant*(6-Ia) *aph*(3′-III)*aadE*	PHEGBS0599	V	ST19
n/a	None	PHEGBS0359	III	ST19
n/a	*erm*(A) *msr*(D) *mef*(A) *tet*(M) *catQ*	PHEGBS0595	V	ST19
ATCC 49619 ^1^	*S. pneumoniae* susceptible control strain			
ATCC 700677	*S. pneumoniae* resistant control for macrolides, penicillin, tetracycline			

^1^ Quality control strains were obtained from the American Type Culture Collection (ATCC).

**Table 2 antibiotics-14-00156-t002:** MIC method concordance by antimicrobial with concordance defined as equal to +/− one dilution of the mode (highlighted in bold).

	MIC Fold Dilution for Agar Versus BMD								
Antimicrobial	−3	−2	−1	0	1	2	3	>+3	% Within ± log_2_ Dilution
Benzylpenicillin			**10**	**36**	**19**	7			90.28
Chloramphenicol		3	**8**	**56**	**5**				95.83
Clindamycin		2	**30**	**34**	**3**	3			93.06
Erythromycin		2	**8**	**31**	**21**	10			83.33
Gentamicin			**4**	**30**	**38**				100.0
Levofloxacin			**3**	**39**	**29**	1			98.61
Tetracycline	3	19	**24**	**9**	**5**	1	5	6	52.78
Vancomycin				**15**	**48**	9			87.50

**Table 3 antibiotics-14-00156-t003:** Concordance and agreement of susceptibility profiles obtained by agar and broth antibiotic susceptibility testing demonstrated by the percentage of repeats with matching susceptibility profiles. Cohen’s kappa statistic given for agreement strength.

Antibiotic	Susceptibility Profiles (Agar-Broth)					% Concordance (SS, II, RR)	Kappa	Agreement Strength
	SS	RS	SR	RR	II			
Chloramphenicol	62		1	9		98.61	0.94	High
Clindamycin	54		1	17		98.61	0.96	High
Erythromycin	22	4		46		94.44	0.88	Moderate
Levofloxacin				9	63	100.00	1	High
Tetracycline	15			57		100.00	1	High

**Table 4 antibiotics-14-00156-t004:** Table showing the dilution series of stock to achieve desired end concentrations for the agar-dilution method.

Primary Stock mg/L	μL of Stock Added to 20 mL	End Concentration mg/L
2560	2000	256
2560	1000	128
2560	500	64
2560	250	32
2560	125	16
2560	62.5	8
80	1000	4
80	500	2
80	250	1
80	125	0.5
80	62.5	0.25
2.5	1000	0.125
2.5	512	0.064
2.5	256	0.032
2.5	128	0.016
2.5	64	0.008

**Table 5 antibiotics-14-00156-t005:** Breakpoints used to establish resistance based on EUCAST guidelines (where available).

Antimicrobial	Sensitive	Increased	Resistant
Benzylpenicillin	≤0.25	-	>0.25
Chloramphenicol *	≤8	-	>8
Clindamycin	≤0.5	-	>0.5
Erythromycin	≤0.25	-	>0.25
Levofloxacin	≤0.001	0.002–2	>2
Gentamicin **			>0.5
Tetracycline	≤1	-	>1
Vancomycin	≤2	-	>2

* EUCAST MIC breakpoints given in mg/L, where available for EUCAST v.14.0. ** Gentamicin given as pK/pD clinical threshold values as MIC breakpoints are not available. Antibiotic test range was 0.008–128 mg/L except for clindamycin (0.008–4 mg/L).

## Data Availability

The data are available from the corresponding author on request. All isolates are available from public culture repositories. ENA references for genomic sequences have been provided for access through public database repositories.

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
