# Peer review of "Agar-Dilution Is Comparable to Broth Dilution for MIC Determination in Streptococcus agalactiae"

_antibiotics, 2025, doi:10.3390/antibiotics14020156_

Round 1
Reviewer 1 Report
Comments and Suggestions for Authors
o Was the stability of antibiotics in agar plates tested over time, particularly for tetracycline, which is known to degrade under certain conditions?
o Why were broth microdilution chosen as the reference standard, and how does it compare with other methods, such as E-test or automated systems like VITEK2, in terms of reproducibility?
o Did the study assess intra-operator variability between agar dilution and broth microdilution methods, especially for antibiotics with high minimum inhibitory concentration (MIC) variability?
o The study used quality control strains. Were additional controls implemented to validate the reproducibility of agar dilution for antibiotics not included in the reference guidelines?
o What factors might explain the lower concordance rates for erythromycin (83.33%) and tetracycline (52.78%) compared to other antibiotics? Could medium composition or the specific properties of the antibiotics play a role?
o The study mentions "trailing growth" as an issue in broth microdilution for erythromycin. Did this phenomenon occur with other antibiotics, and how could agar dilution help mitigate it?
o Gentamicin demonstrated perfect concordance across methods. Is this a characteristic specific to gentamicin's mode of action, or could it indicate a methodological bias that favors its detection?
o Is there evidence in the literature suggesting a systematic bias in tetracycline MIC determination when using broth microdilution compared to agar dilution?
o How do the observed discrepancies in MICs impact clinical decision-making for managing group B Streptococcus (GBS) infections? Would these differences influence resistance interpretation or therapeutic choices?
o Given the high prevalence of tetracycline resistance genes in GBS, should agar dilution still be considered reliable for surveilling tetracycline resistance?
o The manuscript highlights the use of agar dilution for antimicrobial resistance (AMR) surveillance. Can this method be adapted for real-time surveillance, or is it more suitable for retrospective studies due to its batch-processing nature?
o Could modifications, such as using alternative growth supplements or stabilizers, improve the accuracy and concordance of tetracycline and erythromycin MIC determination?
o Is there potential for miniaturizing agar dilution methods (e.g., using microplates) to reduce resource consumption and increase throughput?
o How could the findings of this study contribute to global AMR surveillance programs, especially in regions with limited access to automated systems?
o Would the cost-effectiveness of agar dilution in large-scale surveillance outweigh its resource-intensive setup compared to broth microdilution?
o Should agar dilution be considered for inclusion in EUCAST or CLSI guidelines for specific use cases, and what additional validation would be necessary to achieve this?
o What are the barriers to adopting agar dilution as a recognized method for routine diagnostics?
Author Response
Comment 1. Was the stability of antibiotics in agar plates tested over time, particularly for tetracycline, which is known to degrade under certain conditions?
Response 1: While not part of this project we have examined the progressively increasing MIC for tetracycline in agar-based dilution for tetracycline with Legionella pneumophila AST (doi: 10.1093/jac/dkaa535), however, we have found equivalent loss of tetracycline potency in microbroth dilution as well for Legionella pneumophila (10.1016/j.mimet.2024.107071) with incubation duration, which does not have either serum or lysed blood constituents. It is possible that addition of tetracycline to molten agar at 50oC, or interaction with agar itself, may incur an initial loss of potency, but is unlikely to contribute too much to the inherent instability of tetracycline. We have added this to the discussion.
Comment 2. Why were broth microdilution chosen as the reference standard, and how does it compare with other methods, such as E-test or automated systems like VITEK2, in terms of reproducibility?
Response 2: We would not be able to perform this comparison as we do not have access to the VITEK2 platform. While there are a few published studies utilising VITEK2 for resistance determination only one appears to have performed a comparison and this was only to disk diffusion (Tazi et al., 2007). In addition, EUCAST and CLSI only give guidance for disk diffusion methods (not quantitative) and broth microdilution as the existing “gold standard” methods of AST determination.
Comment 3. Did the study assess intra-operator variability between agar dilution and broth microdilution methods, especially for antibiotics with high minimum inhibitory concentration (MIC) variability?
Response 3: The method was developed by EARP and taught to his student CF. As all plates were read only by these two individuals, and the more junior learned the plate reading from the post-doctoral investigator using this project– it would be inappropriate to try to establish inter-operator variability in this study.
Comment 4. The study used quality control strains. Were additional controls implemented to validate the reproducibility of agar dilution for antibiotics not included in the reference guidelines?
Response 4: Our investigated twenty-four isolate GBS panel were a subset chosen from a larger 195 isolate cohort (doi: 10.1099/mgen.0.000783) that had been fully characterized and genomically sequenced. Isolates were chosen for their genomic determinants (e.g. 5 strains with no known mutations or resistance genes as susceptible controls; catQ – known chloramphenicol resistance; ermA/B, mefA/msrD – known macrolide resistance; ermB, lsaA – known lincosamide resistance; aac(6’)aph(2”) – known high gentamicin MIC; parC/gyrA mutations – known fluoroquinolone resistance; only vancomycin and penicillin resistant isolates were unavailable. From that viewpoint, the investigated GBS cohort were pre-selected for their previously characterised resistance and known resistance mechanism and are themselves deposited in NCTC for anyone who wishes to use them as reference strains in the future.
Comment 5. What factors might explain the lower concordance rates for erythromycin (83.33%) and tetracycline (52.78%) compared to other antibiotics? Could medium composition or the specific properties of the antibiotics play a role?
Response 5: The “trailing” growth was the primary cause of difficulty for erythromycin (establishing an MIC cut-off was subjective and well known). Equally, we have found that tetracycline is often problematic in our previous studies with Legionella as discussed above. There are lots of theories about why tetracycline behaves more variably – cation concentrations, oxidation, etc. As variability exists in agar and broth dilution of multiple compositions more for this antimicrobial than others in our hands, using the same high quality source, and we have investigated the effects of removing/adding magnesium/calcium and antioxidants (both known to influence tetracycline activity) without any significant effect on MICs (unpublished), we are unable to define the underlying cause of this variability for tetracycline.
Comment 6. The study mentions "trailing growth" as an issue in broth microdilution for erythromycin. Did this phenomenon occur with other antibiotics, and how could agar dilution help mitigate it?
Response 6: Yes, as detailed in our manuscript (additionally highlighted in EUCAST guidelines) trailing growth seems particularly bad for erythromycin. We did not observe this phenomenon for the other antimicrobials that we tested. For agar-dilution, there were either colonies on the agar or there were not, so MIC was easy to establish rather than a gradient of turbidity over a few wells where you had to judge at what level the turbidity represented sufficient cessation of growth as it often trailed to some degree to the end of the dilution series.
Comment 7. Gentamicin demonstrated perfect concordance across methods. Is this a characteristic specific to gentamicin's mode of action, or could it indicate a methodological bias that favors its detection?
Response 7: The reviewer has made an astute observation, however, the MIC difference between the single isolate carrying the aac(6’)aph(2”) gene and the remainder of the cohort was fairly substantial. In our experience all GBS strains that carry this gene routinely have MICs >128 mg/L and it is entirely possible that any mechanism of resistance that modifies the antimicrobial, rather than the antimicrobial’s target, would likely yield a change of this magnitude but this would not be biased for any particular methodology. We have added a statement to highlight that in the discussion.
Comment 8. Is there evidence in the literature suggesting a systematic bias in tetracycline MIC determination when using broth microdilution compared to agar dilution?
Response 8: There is no literature suggesting a bias, however, there are numerous reports (including those we have highlighted for Legionella pneumophila from our own lab above), that show tetracycline is labile once dissolved in water and used in any form of AST, and that ferric ion oxidation and magnesium/calcium cations influence tetracycline inhibitory activity.
Comment 9. How do the observed discrepancies in MICs impact clinical decision-making for managing group B Streptococcus (GBS) infections? Would these differences influence resistance interpretation or therapeutic choices?
Response 9: The discrepancies in MIC will not impact clinical decision making, they will only influence future ECOFF thresholds (most notably the ECOFF threshold for erythromycin was decreased from 1.0 to 0.5 mg/L in 2022 by EUCAST to ensure more accurate capture of ermA carrying isolates); patients would not be treated with any of the tetracycline family due to the >90% resistance gene carriage, and the increasing rates of macrolide resistance gene carriage has resulted in macrolide use being discouraged in high income countries as well.
Comment 10. Given the high prevalence of tetracycline resistance genes in GBS, should agar dilution still be considered reliable for surveilling tetracycline resistance?
Response 10: The reviewer raises a valid point, and we have commented on the lack of clinical utility for tetracycline screening in the discussion. As there are EUCAST ECOFF values for tetracycline resistance available, it was included in this investigation for the sake of completeness.
Comment 11. The manuscript highlights the use of agar dilution for antimicrobial resistance (AMR) surveillance. Can this method be adapted for real-time surveillance, or is it more suitable for retrospective studies due to its batch-processing nature?
Response 11: This method could be used for real time surveillance, but only in a context where high daily sample numbers (>40) were being analysed. However, a modified two plate screening method could be implemented (i.e. one plate with no antimicrobials added and one plate containing the ECOFF threshold concentration of antimicrobials – for example, 0.5 mg/L benzylpenicillin – where any growth on the latter would indicate resistance). We have expanded on this in the discussion.
Comment 12. Could modifications, such as using alternative growth supplements or stabilizers, improve the accuracy and concordance of tetracycline and erythromycin MIC determination?
Response 12: As mentioned above with our agar and broth dilution investigations for standardising Legionella pneumophila AST (both published and unpublished), we have investigated the effects of adding calcium and magnesium (identified in the literature as required to bind the ribosome), chelating Fe3+ or using antioxidants (identified in the literature as impacting tetracycline degradation), and comparing MICs in the presence and absence of serum proteins (serum proteins may bind or degrade tetracycline); none of these modifications have made any difference to tetracycline MICs for Legionella pneumophila (and by extrapolation should not affect GBS MICs). To date, no supplements or stabilisers have been identified that would have any effect on this phenomenon. We have added this to the discussion.
Comment 13. Is there potential for miniaturizing agar dilution methods (e.g., using microplates) to reduce resource consumption and increase throughput?
Response 13: Rather than using microplates, we would recommend using growth control (no antimicrobials) and threshold concentrations of antimicrobials as a two-plate method (size of the plate could easily be modified), and restricting testing to clinically relevant antimicrobials (primarily benzylpenicillin, with alternatives of levofloxacin and vancomycin -although second line choice varies for some countries). We proposed this method for Ureaplasma spp. resistance screening in 2009 (doi: 10.1128/AAC.01349-08) and it has subsequently been adopted to development in the widely-used lyophilised commercial assays Mycoplasma-IST3 (doi: doi: 10.1093/jac/dkab320) and Myco Well D-one (doi: 10.1007/s10096-020-03993-7). This has been included in the discussion.
Comment 14. How could the findings of this study contribute to global AMR surveillance programs, especially in regions with limited access to automated systems?
Response 14: As outlined above, use of a growth control plate and a single antimicrobial plate containing the resistance threshold concentration for each antimicrobial, could be implemented easily and spotting 1 microliter of the McFarland standard suspension for each isolate could be used in place of using a multipin inoculator. This has been included in the discussion. However, LMIC resources would have difficulty in accessing a commercial source of lysed horse blood with cold-chain delivery and beta-NAD, which would be required for any method including disk diffusion. However, any facility capable of performing disk diffusion could also utilise the adapted two-plate method we have now included in the discussion.
Comment 15. Would the cost-effectiveness of agar dilution in large-scale surveillance outweigh its resource-intensive setup compared to broth microdilution?
Response 15: Absolutely, at large scale with high-throughput the cost of the agar dilution is much lower than broth dilution for plastic plate consumption (96 isolates would require twelve 96-well plates (only 8 isolates can be read per BMD plate) relative to eight plates for full MIC range or 2 if using threshold concentrations for agar dilution. Furthermore, the staff time required to set up the dilution series for those twelve 96-well plates far exceeds the time (and potential for error risk) relative to the eight (or 2 for threshold) plates for agar dilution. An added benefit to the agar dilution method is that presence of non-GBS colonies (size, colour, texture) would be very apparent compared to the other isolates on the plate, whereas turbidity does not give any indication of the bacterial characteristics generally.
Comment 16. Should agar dilution be considered for inclusion in EUCAST or CLSI guidelines for specific use cases, and what additional validation would be necessary to achieve this?
Response 16: We are hopeful that this publication will start that process (at least for EUCAST), I believe CLSI guidelines have to be led by a U.S. based reference laboratory. For both of these governing bodies, additional validation would require this method to be repeated using defined sources of culture components and antimicrobials across a number of reference laboratories, with analysis of inter- and intra-laboratory variation as part of validation before being adopted in official guidelines.
Comment 17. What are the barriers to adopting agar dilution as a recognized method for routine diagnostics?
Response 17:
- Sample size: The use of a multipin inoculator would only be justified for batches of >40 isolates. For 8 or fewer isolates, investigated for only one or two antimicrobials BMD would be more suitable, with the exception of employing the two-plate resistance threshold method for each antimicrobial outlined above.
- Rate: If several isolates need to be tested daily as routine, agar plates (again particularly if using the two-plate resistance threshold variation) store better and longer, with less issue of evaporation and tipping, than plates containing broth dilution. Where as if 8 or fewer isolates are only investigated once or twice a week, BMD would be more suitable.
- Range of antimicrobials: Even testing 8 isolates against 8 antimicrobials daily would require considerable set-up and resources, whereas multipin inoculators can be configured to inoculate up to 20 isolates on 60 mm diameter plates, and pouring replicate series of antimicrobial dilutions in agar will store better and represent less set up time if performed once or twice a month for daily investigations of an array of antimicrobials.
Reviewer 2 Report
Comments and Suggestions for Authors
Several comments for improvement:
1. Add detailed methodological descriptions for both agar dilution and MIC techniques, particularly for steps that might influence variability in MIC results (e.g., the handling of lysed horse blood and β-NAD supplements). This would enhance reproducibility for readers.
2. The statistical methodology, particularly the calculation of Cohen’s kappa values, should be explained in more detail.
3. Please enrich the discussion on how agar-dilution could be applied to detect and track novel resistance mechanisms in S. agalactiae and other pathogens.
4. The concordance for tetracycline results is quite low (52.78%), compared to other antibiotics. Line 162-165, This part need more discussion on factor that contributes to low concordance (e.g. stability of tetracycline in storage and preparation condition, different medium, maybe due to also extended incubation?. Suggestion also that could be added in discussion for cross referencing with alternative method such as E-test for validation.
5. Add more discussion on the interpretation of Cohen’s kappa values, emphasizing what constitutes acceptable agreement for clinical and surveillance purposes. Discuss whether the observed values for tetracycline are acceptable for AMR studies.
6. Discuss whether agar-dilution could be extended beyond GBS to other pathogens or antibiotic classes, potentially broadening its applicability for AMR testing in general microbiology labs.
Author Response
Comment 1. Add detailed methodological descriptions for both agar dilution and MIC techniques, particularly for steps that might influence variability in MIC results (e.g., the handling of lysed horse blood and β-NAD supplements). This would enhance reproducibility for readers.
Response 1: We thank the reviewer for pointing out this oversight. We have expanded the methods section to enable readers to reproduce our methods.
Comment 2. The statistical methodology, particularly the calculation of Cohen’s kappa values, should be explained in more detail.
Response 2: We have added a statistical analyses section to the methods and explained the statistical interrogation more thoroughly in the results where they are utilised. Whilst most commonly used for measures of inter-rater reliability, Cohen’s kappa can be applied to measure the reliability of two different methodologies, with results collected by one individual, over a series of experimental repeats. This study utilised Cohen’s kappa to measure the agreement between antibiotic susceptibility profiles obtained by both broth microdilution and agar-dilution methodologies, with susceptibility profile thresholds defined by EUCAST. Intra-rater data was categorised as susceptible, intermediate, or resistant, with results collected from the three experimental repeats compared by weighted kappa. Calculation of the kappa value utilised the GraphPad online calculator (Quantify agreement with kappa). Interpretation of the kappa statistic followed the categories defined by Landis and Koch (1977). As Cohen’s kappa was measuring the agreement of categorical groupings, percentage agreement of MIC values, agreement considered ± log2 dilutions, was calculated to analyse the individual data points.
Comment 3. Please enrich the discussion on how agar-dilution could be applied to detect and track novel resistance mechanisms in S. agalactiae and other pathogens.
Response 3: We have included reference to our work using agar dilution in Legionella pneumophila, however, agar dilution methods are used for many other Gram negative organisms routinely. This method does not track resistance mechanisms, only phenotypes, so would be unsuitable to suggest it could track mechanisms unless combined with whole genome sequencing.
Comment 4. The concordance for tetracycline results is quite low (52.78%), compared to other antibiotics. Line 162-165, This part need more discussion on factor that contributes to low concordance (e.g. stability of tetracycline in storage and preparation condition, different medium, maybe due to also extended incubation?. Suggestion also that could be added in discussion for cross referencing with alternative method such as E-test for validation.
Response 4: Whilst correct that the concordance with tetracyclines was low, it is worth noting that there was 100% concordance between calling of resistant and sensitive isolates. We have included more discussion discussing our experience with tetracycline AST with Legionella pneumophila, which included validation of E-strips (that showed worse correlation to broth than agar dilution). We have also included, in response to Reviewer 1’s comments, that tetracycline surveillance is of limited utility given the prevalence of >90% tetracycline resistance gene carriage.
Comment 5. Add more discussion on the interpretation of Cohen’s kappa values, emphasizing what constitutes acceptable agreement for clinical and surveillance purposes. Discuss whether the observed values for tetracycline are acceptable for AMR studies.
Response 5: We have added more discussion of the difference between kappa values and concordance as requested by the reviewer.
Comment 6. Discuss whether agar-dilution could be extended beyond GBS to other pathogens or antibiotic classes, potentially broadening its applicability for AMR testing in general microbiology labs.
Response 6: This is an interesting point, and the wider use of agar dilution has been mentioned as above, additional work on agar dilutions use in non GBS species has been mentioned and extension to other antimicrobials would be contingent on being able to compare isolates with and without defined resistance genes or resistance-mediating mutations to ensure that the MICs accurately match phenotype and genotype. It is our hope that that this approach will extend to other Gram-positive organisms. As referenced, it has previously been performed for S. pyogenes.
Reviewer 3 Report
Comments and Suggestions for Authors
The manuscript is an original study. However, the following points need to be revised or taken into account:
Line 40: “Group B Streptococcus (GBS)”: Streptococcusshould be italicized.
Line 41: “E.coli” should be written as Escherichia coli.
Line 54-56: “Neonatal treatment... not recommended.”: References should be given for these expressions.
Line 68-75: Passive voice (sentences) should be used. In addition, explanations based on reference no. 14 should be stated more clearly. The purpose should be written in the last part of this paragraph. ,
Line 78: The parentheses at the end of the sentence should be deleted.
Line 1341. “All data points performed in triplicate.”. This statement must be deleted. It should be given in the method section.
Line 155: “Amsler et al. [19]”.
Line 157; Line 174: Passive voice is used as the appropriate language in academic studies. Please take this into consideration. Expressions such as “In this study or in the current/present study” can be used.
Line 190: DMSO?
Line 191: “these” The sentence must start with a capital letter.
Line 194: The name of the table must be descriptive.
Line 196: “MH agar” The company and country information for this medium should be given.
The discussion section can be enriched with new sources or the given references can be detailed.
Author Response
The manuscript is an original study. However, the following points need to be revised or taken into account:
Comment 1. Line 40: “Group B Streptococcus (GBS)”: Streptococcus should be italicized.
Response 1: This has now been italicised.
Comment 2: Line 41: “E.coli” should be written as Escherichia coli.
Response 2: This has been amended.
Comment 3: Line 54-56: “Neonatal treatment... not recommended.”: References should be given for these expressions.
Response 3: We have now added relevant references.
Comment 4: Line 68-75: Passive voice (sentences) should be used. In addition, explanations based on reference no. 14 should be stated more clearly. The purpose should be written in the last part of this paragraph.
Response 4: This has been amended.
Comment 5: Line 78: The parentheses at the end of the sentence should be deleted.
Response 5: This has been amended.
Comment 6: Line 1341. “All data points performed in triplicate.”. This statement must be deleted. It should be given in the method section.
Response 6: This been moved to the methods
Comment 7: Line 155: “Amsler et al. [19]”.
Response 7: This has been amended.
Comment 8: Line 157; Line 174: Passive voice is used as the appropriate language in academic studies. Please take this into consideration. Expressions such as “In this study or in the current/present study” can be used.
Response 8: This has been amended.
Comment 9: Line 190: DMSO?
Response 9: DMSO has been defined in full as dimethyl sulfoxide This has been amended.
Comment 10: Line 191: “these” The sentence must start with a capital letter.
Response 10: This has been amended.
Comment 11: Line 194: The name of the table must be descriptive.
Response 11:This has been amended.
Comment 12: Line 196: “MH agar” The company and country information for this medium should be given.
Response 12: the supplier has been defined.
Comment 13: The discussion section can be enriched with new sources or the given references can be detailed.
Response 13: The discussion has been expanded and diversified, with more references given as requested.
Round 2
Reviewer 1 Report
Comments and Suggestions for Authors
Thanks for the clarification.